# PATCHED DENOISING DIFFUSION MODELS FOR HIGH-RESOLUTION IMAGE SYNTHESIS

**Zheng Ding**[1]* **Mengqi Zhang**[1]* **Jiajun Wu**[2] **Zhuowen Tu**[1]
[1]UC San Diego [2] Stanford University

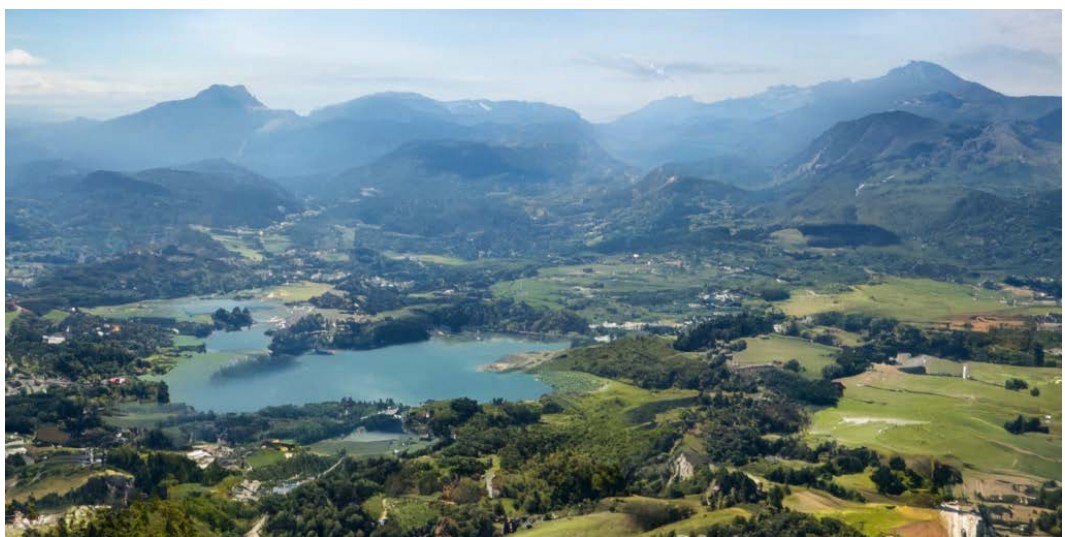

Figure 1: Generated image of size 1024×512 using the model trained on 21k natural images using a 148M-parameters model.

## ABSTRACT

We propose an effective denoising diffusion model for generating high-resolution images (e.g., 1024×512), trained on small-size image patches (e.g., 64×64). We name our algorithm Patch-DM, in which a new feature collage strategy is designed to avoid the boundary artifact when synthesizing large-size images. Feature collage systematically crops and combines partial features of the neighboring patches to predict the features of a shifted image patch, allowing the seamless generation of the entire image due to the overlap in the patch feature space. Patch-DM produces high-quality image synthesis results on our newly collected dataset of nature images (1024×512), as well as on standard benchmarks of LHQ(1024× 1024), FFHQ(1024× 1024) and on other datasets with smaller sizes (256×256), including LSUN-Bedroom, LSUN-Church, and FFHQ. We compare our method with previous patch-based generation methods and achieve state-of-the-art FID scores on all six datasets. Further, Patch-DM also reduces memory complexity compared to the classic diffusion models. Project page: https://patchdm.github.io.

## 1 INTRODUCTION

There have been explosive developments in generative adversarial learning (Tu, 2007; Goodfellow et al., 2014; Radford et al., 2016; Arjovsky et al., 2017; Karras et al., 2019; Donahue et al., 2017), though many GAN models remain hard to train. VAE models (Kingma & Welling, 2014) are easier to train, but the resulting image quality is often blurry. Diffusion generative models have lately gained tremendous popularity with generated images of superb quality (Sohl-Dickstein et al., 2015; Ho et al., 2020; Song & Ermon, 2020; Song et al., 2020; Croitoru et al., 2023; Ramesh et al., 2022). Despite the excellent modeling capability of generative diffusion models, the current models still face challenges in both training and synthesis.

---

*Equal Contribution.

Due to direct optimization in the pixel space and multi-timestep training and inference, diffusion models are hard to scale up to high-resolution image generation. Therefore, current state-of-the-art models either use super-resolution methods to increase the generated images to higher resolutions (Ramesh et al., 2022; Saharia et al., 2022a), or optimize the latent space instead of the pixel space (Rombach et al., 2022). However, both types of approaches still consist of high-resolution image generators that consume a large memory with a big model size.

To ameliorate the limitations in the current diffusion models, we propose a new method, Patch-DM, to generate high-resolution images with a newly-introduced feature collage strategy. The basic operating point for Patch-DM is a patch-level model that is relatively compact compared to those modeling the entire image. Though it appears to have introduced compromises for a patch-based representation, Patch-DM can perform seamless full-size high-resolution image synthesis without artifacts of the boundary effects for pixels near the borders of the image patches. The effectiveness of Patch-DM in directly generating high-resolution images is enabled by a novel feature collage strategy. This strategy helps feature sharing by implementing a sliding-window based shifted image patch generation process, ensuring consistency across neighboring image patches; this is a key design in our proposed Patch-DM method to alleviate the boundary artifacts without requiring additional parameters. To summarize, the contributions of our work are listed as follows:

- We develop a new denoising diffusion model based on patches, Patch-DM, to generate images of high-resolutions. Patch-DM can perform direct high-resolution image synthesis without introducing boundary artifacts.
- We design a new feature collage strategy where each image patch to be synthesized obtains features partially from its shifted input patch. Through systematic window sliding, the entire image is being synthesized by forcing feature consistency across neighboring patches. This strategy, named feature collage, gives rise to a compact model of Patch-DM that is patch-based for high-resolution image generation.

Patch-DM points to a promising direction for generative diffusion modeling at a flexible patch-based representation level, which allows high-resolution image synthesis with lightweight models.

## 2  RELATED WORK

**Generative diffusion models.**    Generative diffusion models (Sohl-Dickstein et al., 2015; Ho et al., 2020; Song & Ermon, 2020) which learn to denoise noisy images into real images have gained much attention lately due to its training stability and high image quality. Lots of progress has been made in diffusion models such as faster sampling(Song et al., 2021), conditional generation(Nichol & Dhariwal, 2021; Dhariwal & Nichol, 2021) or high-resolution image synthesis(Rombach et al., 2022). The traits of diffusion models have been amplified particularly by the success of DALL·E 2 (Ramesh et al., 2022) and Imagen (Saharia et al., 2022a) which generate high quality images from the given texts.

**Patch-based image synthesis.**    The practice of employing image patches of relatively small sizes to generate images of larger sizes has been a longstanding technique in computer vision and graphics, particularly in the context of exemplar-based texture synthesis (Efros & Leung, 1999). While generative adversarial networks (GANs) have been utilized for expanding non-stationary textures(Zhou et al., 2018), image synthesis is still considered more challenging due to the complex structures present in images. To address this challenge, COCO-GAN(Lin et al., 2019) uses micro coordinates and latent vectors to synthesize large images by generating small patches first. InfinityGAN(Lin et al., 2022) further improves this by introducing Structure Synthesizer and Padding Free Generator to disentangle global structures and local textures and also generate consistent pixel values at the same spatial locations. ALIS(Skorokhodov et al., 2021) proposes an alignment mechanism on latent and image space to generate larger images. Anyres-GAN(Chai et al., 2022), on the other hand, adopts a two-stage training method by first learning the global information from low-resolution downsampled images and then learning the detailed information from small patches. There are also some works share similar directions on patch-based diffusion models. Luhman & Luhman (2022) does a reshaping operation on the input image which pushes the dimensions of the height and width to the channels. The model still takes the whole image as input just with the shape changed. Wang et al. (2023) does a patch operation during the training stage by concatenating another position embedding layer to the input while still requires full-resolution during the inference

stage. There are also other works applying patch-based diffusion models to specific applications like image restoration under weather conditions(Özdenizci & Legenstein, 2023) and anomaly detection in brain MRI(Behrendt et al., 2024). Both of them utilize a conditional diffusion mechanism by utilizing the weather-degraded images or images that miss some patches as conditions.

Our work, Patch-DM consists of a new design, feature collage, in which partial features of neighboring patches are cropped and combined for predicting a shifted patch. We borrow the term "collage" from the picture collage task (Wang et al., 2006) for the ease of understanding of our method, though our feature collage strategy only has a loose conceptual connection to picture collage (Wang et al., 2006). Adopting positional embedding in Patch-DM also makes it easier to maintain spatial regularity. Although Patch-DM employs a shifted window strategy, its motivation and implementation are different from those of the widely-known Swin Transformers (Liu et al., 2021).

## 3 BACKGROUND

Denoising diffusion models generate real images from randomly sampled noise images by learning a denoising function (Ho et al., 2020). Instead of directly denoising the random noise image to a real image, denoising diffusion models learn to denoise the noise image through $T$ steps. The forward process adds noise to the image $x_0$ gradually while the learned denoising function $f_\theta$ tries to reverse this process from the $x_T \sim \mathcal{N}(0, \mathbf{I})$. More formally, the forward process at time step $t(t = 1...T)$ can be defined as

$$x_t \sim \mathcal{N}(x_{t-1}; \sqrt{1 - \beta_t} x_{t-1}, \beta_t \mathbf{I}), \tag{1}$$

where $\beta_t$ are hyperparameters that control the noise, making the noise level of $x_t$ gradually larger through the timesteps. Note that $x_t$ can be directly derived from the original image $x_0$ since Eq. 1 can be rewritten as

$$x_t \sim \mathcal{N}(x_0; \sqrt{\alpha_t} x_0, (1 - \alpha_t)\mathbf{I}), \tag{2}$$

where $\alpha_t = \prod_{s=1}^{t}(1 - \beta_s)$. In order to generate the images from the noise input, the denoising model $f_\theta$ learns to reverse from $x_t$ to $x_{t-1}$, which is defined as

$$\hat{\epsilon}_t = f_\theta(x_t, t), \tag{3}$$

$$x_{t-1} \sim \mathcal{N}(x_t; \frac{1}{\sqrt{1 - \beta_t}}(x_t - \frac{\beta_t}{\sqrt{1 - \alpha_t}}\hat{\epsilon}_t), \sigma_t \mathbf{I}), \tag{4}$$

where $\sigma_t$ are hyperparameters that control the variance of the denoising process. The objective of the denoising model is $||\epsilon_t - \hat{\epsilon}_t||^2$. $\epsilon_t$ is ground truth noise added on image.

Therefore, after the denoising model is trained, the model can generate real images from random noise using Eq. 4. As can be seen, the whole generation process depends fully on the denoising process. Since the model denoises the image in the pixel space directly, the computation would be very expensive once the resolution gets higher.

## 4 PATCHED DENOISING DIFFUSION MODEL

In this section, we describe our proposed Patched Denoising Diffusion Model (Patch-DM). Rather than using entire complete images for training, our model only takes patches for training and inference, and uses our proposed feature collage mechanism to systematically combine partial features of neighboring patches. Consequently, Patch-DM is capable of resolving the issue of high computational costs associated with generating high-resolution images, as it is resolution-agnostic.

Before we dive into our model's training details, we first give an overview of the image generation process of our method. The training image from the dataset is $x_0 \in \mathbf{R}^{C \times H \times W}$, we split $x_0$ into $x_0^{(i,j)}$ where $i, j$ is the row and column number of the patch, $x_0^{(i,j)} \in \mathbf{R}^{C \times h \times w}$. Instead of directly generating $x_0$ like most of methods do, our model only generates $x_0^{(i,j)}$ and concatenate them together to form a complete image.

A very basic way to do this is what we show in Figure 2(a) where the denoising model takes the noised image patch $x_t^{(i,j)}$ as input and output the corresponding noise $\hat{\epsilon}_t^{(i,j)}$. However, since the patches do not interact with each other, there will be severe borderline artifacts.

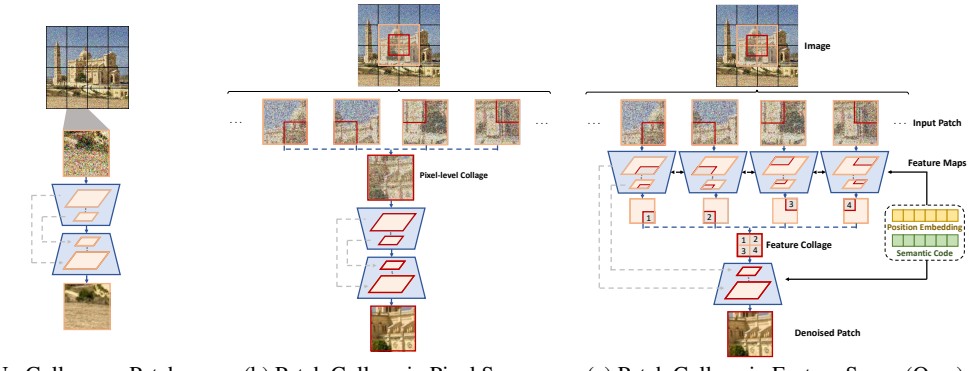

(a) No Collage on Patches      (b) Patch Collage in Pixel Space      (c) Patch Collage in Feature Space (Ours)

Figure 2: **Patch Generation For Image Synthesis.** (a) shows a very basic method of patch-wise image synthesis by simply splitting the images and generating patches independently. This method brings severe border artifacts. (b) alleviates the border artifacts by using shifted windows while generating images and doing patch collage in pixel space. (c) is our proposed method which collages the patches in the feature space. The features for neighboring features will be split and collaged for a new patch synthesis. We will show this method is a key design for us to generate high-quality images without border artifacts.

A further way to do this is to shift image patches during each time step depicted in Figure 2(b). At different time steps, the model will take either the original split patch $x_t^{(i,j)}$ or the shifted split patch $x_t'^{(i,j)}$ so that the border artifacts can be alleviated which we call "Patch Collage in Pixel Space". However, in Section 6 we show that the border artifacts still exist.

To further improve this method, we propose a novel feature collage mechanism depicted in Figure 2(c). Instead of performing patch collage in the pixel space, we perform it in the feature space. The patch collage in the feature space is more in-depth and supports multi-level interaction. This allows the patches to be more cognizant of the adjacent features and prevent border artifacts from appearing while generating the complete images. More formally,

$$[z_1^{(i,j)}, z_2^{(i,j)}, ..., z_n^{(i,j)}] = f_\theta^E(x_t^{(i,j)}, t), \tag{5}$$

where $f_\theta^E$ is the UNet encoder and $z_1^{(i,j)}, z_2^{(i,j)}, ..., z_n^{(i,j)}$ are the internal feature maps. We then split the feature maps and collage the split feature maps to generate shift patches

$$\begin{aligned}\hat{z}_k'^{(i,j)} = [&P_1(z_k^{(i,j)}), P_2(z_k^{(i,j+1)}), \\ &P_3(z_k^{(i+1,j)}), P_4(z_k^{(i+1,j+1)})],\end{aligned} \tag{6}$$

where $P1, P2, P3, P4$ are split functions as shown in Figure 2(c). Then we send these collaged shift features $\hat{z}_k'^{(i,j)}$ to the UNet decoder to get the predicted shift patch noise:

$$\epsilon_t'^{(i,j)} = f_\theta^D([z_1'^{(i,j)}, z_2'^{(i,j)}, ..., z_n'^{(i,j)}], t). \tag{7}$$

In order to make the model generate more semantically consistent images, we also add position embedding and semantic embedding to the model so that $f_\theta$ will take another two inputs which are $\mathcal{P}(i,j)$ and $\mathcal{E}(x_0)$.

During inference time, we take a 3x3 example as illustrated in Figure 3, in order to generate the border patches, we first pad the images so that the feature collage can be done for each patch without information loss. At each time step $t$, image $x_t$ is decomposed into patches which are fed into the subsequent encoder. Before a feature map goes through the decoder, a split and collage operation is applied to it. Thus, the decoder outputs the predicted noise of the shifted patch. According to Eq. 4, we are able to obtain $x_{t-1}$ and thus generate the final complete images.

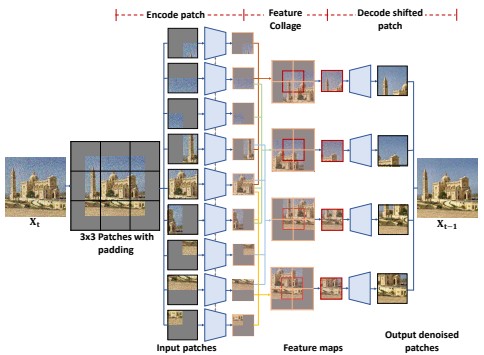

Figure 3: Detailed Inference process at each time step.

Table 1: Quantitative comparison with previous patch-based image generation methods. All models are trained on the natural images dataset (1024×512), standard benchmarks LHQ(1024×1024) and FFHQ(1024×1024). We use FID to measure the overall quality of generated images and sFID for the quality of high-level structures.

| Method | Patch Size | Nature-21K(1024×512) | | LHQ(1024×1024) | | FFHQ(1024×1024) | |
| --- | --- | --- | --- | --- | --- | --- | --- |
| | | FID | sFID | FID | sFID | FID | sFID |
| COCO-GAN(Lin et al., 2019) | 64×64 | 70.980 | 74.208 | 35.693 | 88.988 | 80.059 | 209.683 |
| InfinityGAN(Lin et al., 2022) | 101×101 | 46.550 | 70.041 | 50.646 | 92.577 | 174.789 | 270.699 |
| Anyres-GAN(Chai et al., 2022) | 64×64 | 44.173 | 34.430 | 130.591 | 100.041 | 67.076 | 170.911 |
| Patch-DM (Ours) | 64×64 | **20.369** | **34.405** | **23.777** | **37.217** | **19.696** | **36.512** |

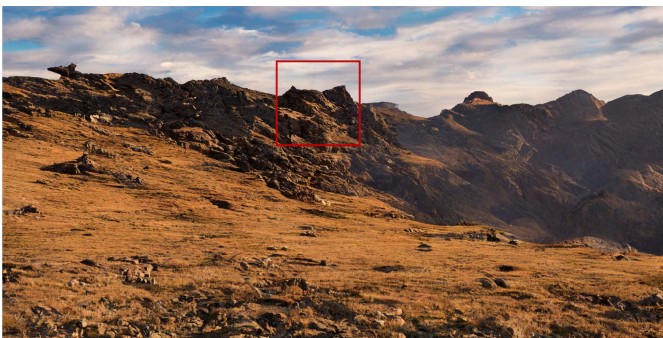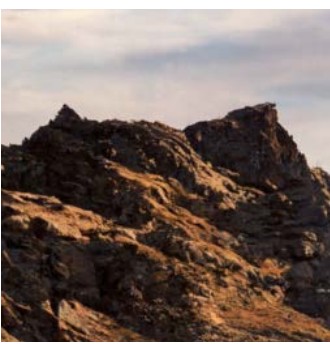

Figure 4: Generated 2048×1024 image. We double the number of patches so that the model can generate images with 2x resolution from 1024×512. The left image is a 2048×1024 image, and the right image is a zoom-in of the red bounding box, with a resolution of 256×256.

## 5 EXPERIMENTS

### 5.1 IMPLEMENTATION DETAILS

**Architecture.** For the model architecture, we base our denoising U-Net model from (Dhariwal & Nichol, 2021) with changes of taking global conditions and positional embeddings. We use two methods to obtain the global conditions. The first is to use a pretrained model to obtain the image features and use the image features as the global conditions, while optimizing the features directly during training. In this case, we do not have to increase the model parameters and can scale to high-resolution images. However, when the number of images in the training dataset is too large, optimizing the pre-obtained image features requires more effort. We use this approach for global conditioning when training on datasets of 1024×512 images. The pretrained model we use for obtaining the image embeddings is CLIP(Radford et al., 2021). We resize the images to 224×224 and send them to ViT-B/16 to obtain the features as global conditions; we then optimize these global conditions directly.

The second is jointly training an image encoder and using its output as the global conditions. Here, the jointly training image encoder may borrow the same architecture as in the denoising U-Net's encoder. It works particularly well when the training dataset is large. However, it requires another model, which would be a bottleneck in training on high-resolution datasets, since the computation would increase significantly as the resolution increases. We use this approach when training on large datasets of 256×256 images. We utilize global conditions with a dimension of 512 in both methods.

**Classifier-free guidance.** We also use the classifier-free guidance (Ho & Salimans, 2022) to improve the training speed and quality. We use classifier-free guidance on both the global conditions and position embeddings. The dropout rate is 0.1 for the global conditions and 0.5 for position embeddings.

**Patch size.** We use a patch size of 64×64 in all our experiments. The denoising U-Net model's architecture is the same across all the datasets, as it is only related to the patch size regardless of the training images' resolution.

**Inference.** Once the denoising U-Net model has been trained, we train another latent diffusion model for unconditional image synthesis. The latent diffusion model's architecture is based on the

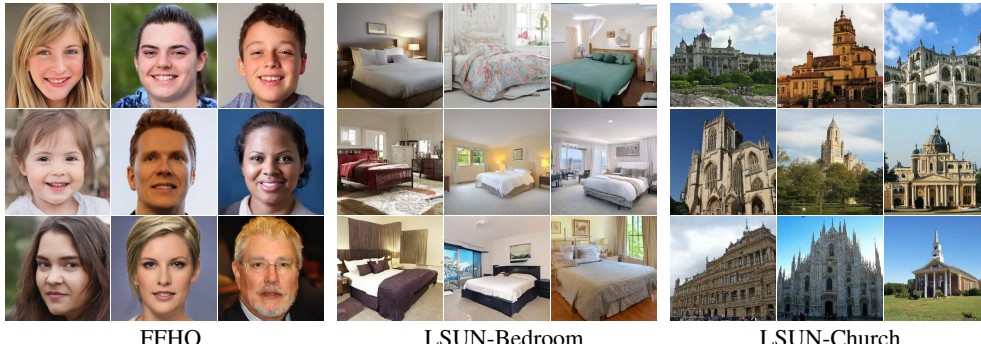

FFHQ                LSUN-Bedroom                LSUN-Church

Figure 5: Generated images on FFHQ, LSUN-Bedroom, and LSUN-Church datasets using our proposed method. All the resolutions are 256×256.

one described in (Preechakul et al., 2022). The data we use for training the latent diffusion model is either from the output of the trained image encoder or the directly optimized image embeddings. To synthesize an image, we will first sample a latent code from the latent diffusion model and then use this latent code to serve as the global conditions for sampling an image. During the sampling stage, we use the inference process proposed by DDIM (Song et al., 2021) and set the sampling step to 50.

**Evaluation.** We conduct both qualitative and quantitative evaluations on four datasets. For quantitative evaluation, we use FID, a popular metric in generative modeling(Heusel et al., 2017) and sFID(Nash et al., 2021) for the quality of high-level structures. To compute FID, we follow the setting of (Heusel et al., 2017) and generate 50K images to compute the metrics over the full dataset. We apply the same setting for sFID.

## 5.2 RESULTS ON 1k-RESOLUTION IMAGES

**Setup.** To show our model's capability on direct high-resolution image synthesis, we show our model's performance on three datasets: LHQ(1024× 1024), FFHQ(1024×1024) and a self-collected 21443 natural images from Wallpaperscraft (2013-2023)(1024×512). We use a patch size of 64×64 across all datasets. Therefore, each image is split into 16×16 for 1024×1024-resolution and 16×8 for 1024×512-resolution.

We use the CLIP ViT-B/16 pretrained image visual encoder to obtain the image embeddings first. While downsampling to fit the CLIP model may result in the loss of some detailed image information in the embeddings, these details can still be "recovered" during the embedding optimization in the training process, aided by the supervision of the original high-resolution image. Each image patch can be trained and sampled independently with feature collage assisting to be aware of the surrounding information. Since every image is segmented into smaller patches, the total number of model parameters is much smaller than other large diffusion models.

As most existing diffusion models merely can directly generate images of 1k resolution, and the general strategy for high-resolution synthesis is to sample hierarchically (generate relatively low-resolution images first and then perform super-resolution), our Patch-DM simplifies sample procedure using much more lightweight models, which is one of the main advantages.

**Results.** We compare our model with previous patch-based image generation methods in Table 1. From the table, we can see our method delivers the best overall quality of the generated images on both FID and sFID scores. Apart from the quantitative evaluation, we also present an image generated by our model in Figure 1. For more generated images, please refer to our supplementary material.

## 5.3 RESULTS ON 256×256 IMAGES

**Setup.** To compare with other existing generative models, we also train our Patch-DM on three standard public datasets: FFHQ, LSUN-Bedroom, and LSUN-Church, and evaluate its sampling performance. All the resolution is 256×256. Thus, the number of patches is 4×4. Notice that the

model architecture keeps the same; the only change here is the number of patches during training and inference. We use the same training setting across the three datasets.

**Results.** We report the quantitative results in Table 2 and qualitative results in Figure 5, respectively. In Table 2, we can see our model achieves competitive results while still outperforms previous patch-based methods. Figure 5 illustrates that despite producing small image patches, our denoising model exhibits minimal boundary artifacts and offers good visual quality. This demonstrates the effectiveness of our feature collage mechanism.

Table 2: Evaluation Metrics of unconditional image synthesis on three 256×256 datasets: FFHQ, LSUN-Bedroom, and LSUN-Church. For a fair comparison, results are reproduced in the same sampling steps as ours i.e. 50, using provided pretrained checkpoints of other diffusion models. We adopt a patch size of 64×64 for Patch-DM, Anyres-GAN, COCO-GAN and 101×101 for InfinityGAN. We bold the numbers to denote the best numbers in the same category (top: non-patch-based methods, bottom:patch-based methods).

| | FFHQ | | LSUN-Bedroom | | LSUN-Church | |
|---|---|---|---|---|---|---|
| Method | FID | sFID | FID | sFID | FID | sFID |
| LDM Rombach et al. (2022) | 8.76 | **7.09** | 3.40 | 7.53 | 4.23 | 11.44 |
| UDM Kim et al. (2022) | **5.54** | - | 4.57 | - | - | - |
| DiffAE Preechakul et al. (2022) | 9.71 | 10.24 | - | - | - | - |
| PGGAN Karras et al. (2018) | - | - | 8.34 | 9.21 | 6.42 | **10.48** |
| StyleGAN Karras et al. (2018) | - | - | **2.35** | **6.62** | **4.21** | - |
| COCO-GANLin et al. (2019) | 34.02 | 37.44 | 41.84 | 62.69 | 17.91 | 73.94 |
| InfinityGAN Lin et al. (2022) | 28.87 | 127.92 | 10.71 | 19.28 | 7.08 | 33.58 |
| Anyres-GAN Chai et al. (2022) | 24.48 | 55.77 | 15.65 | 56.24 | 17.09 | 80.66 |
| Patch-DM (Ours) | **10.02** | **10.58** | **6.04** | **9.93** | **5.49** | **14.80** |

**Model size comparison.** Compared with other widely used diffusion models, our proposed method could achieve competitive performance using a smaller model with above mentioned indispensable components. Patch-DM is fully built upon the network on 64×64 patches regardless of the target image resolution and uses optimized global conditions to avoid the increase of model parameter amounts brought by higher input resolution. Comparison of model parameters with other classic diffusion models on 256×256 resolution is shown in Table 3. Notice that we use the same diffusion model architecture for the 1024×1024 and the 1024×512 resolutions.

Table 3: Number of parameters comparison between different diffusion models on 256×256 resolution. SE means semantic encoder to extract global information. The size of our previously trained 1024×1024 and 1024×512 models is 70M+[size of optimized semantic embeddings] during training and [63M latent DPM] in inference.

| Method | Model Size ↓ |
|---|---|
| **Base model + Super-resolution** | |
| SR3 Saharia et al. (2022b) | B[64]+625M |
| **Direct generation** | |
| ADM Dhariwal & Nichol (2021) | 552M |
| DiffAE Preechakul et al. (2022) | 232M |
| LDM Rombach et al. (2022) | 274M |
| Patch-DM (Ours, full model) | 154M |
| Patch-DM (Ours, w/ SE, w/o latent DPM) | 91M |
| Patch-DM (Ours, w/o SE, w/o latent DPM) | 70M |

## 5.4 APPLICATIONS

We now demonstrate several applications of our Patch-DM. All of them are conducted without post-training.

**Beyond patch generation.** Since our method samples images using patches, we have the option to incorporate more patches during testing. This enables the model to produce images with higher resolutions compared to the ones in the training set without requiring further training. We adopt two ways to achieve this.

The first one is to add more patches internally. We test this on our Nature-21K dataset. To generate 512×1024 images which has the same resolution as the training dataset, we need 8×16 patches (random gaussian noise maps) to start with. To generate a 2× resolution images, we can add another 8×16 patches internally so that the total patch number will become 16×32 that can generate images

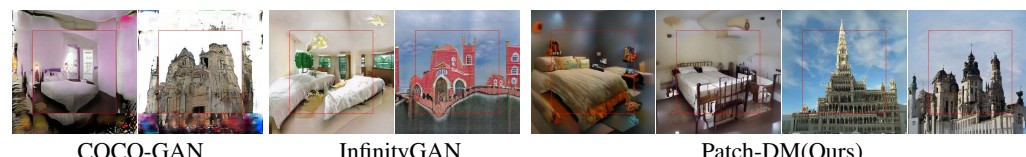

|  COCO-GAN | InfinityGAN | Patch-DM(Ours) |

Figure 6: **Synthesized 384×384 images on LSUN-Bedroom (256×256) and LSUN-Church (256×256).**The left half of each group is LSUN-Bedroom and the right half of each group is LSUN-Church. Despite only being trained on 256×256 images, our model can generate 384×384 images by adding more patches (outside the red bounding box). Extended images generated by our models are compared with COCO-GAN and InfinityGAN, which also have the ability to extend fields without further training.

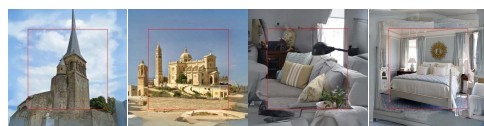

Figure 7: **Image outpainting on LSUN-Church and LSUN-Bedroom.** The image inside the red bounding box is the input image from the validation dataset. We pad the image patches from 4×4 to 6×6 to enable the image outpainting. The image parts outside the red bounding box are the outpainting results.

with a resolution of 1024×2048. The global conditions we use for these patches is the same as the original, while we interpolate the position embeddings in this setting. We provide a generated 2048×1024 image in Figure 4. As can be seen, our model can still generate consistent patches even though the newly added patches have never been used in the training process.

The second one is to add patches outside the original image. This way is similar to beyond-boundary generation in COCO-GAN(Lin et al., 2019) with a key difference that it needs a post-training process to improve the continuity among patches. We conduct an experiment on LSUN-Bedroom and LSUN-Church by adding more patches. The original resolution in the training data we use is 256×256, which is divided by 4×4 patches. We add more patches to the existing 4×4 patches so that the number grows to 6×6. Therefore, the model can generate an image with a resolution of 384×384. For the original 4×4 patches, we use the condition generated from the latent diffusion model and the position embeddings as pre-defined. For the additional patches, we use the same semantic condition, while we don't use position embeddings for these added ones as in this case we're adding patches outside which the position embeddings could not be interpolated. Thus the model needs to synthesize the additional patches only according to the global conditions and the neighboring context information. We present our results in Figure 6 and compare it with COCO-GAN(Lin et al., 2019) without post-training and InfinityGAN(Lin et al., 2022) under the same setting.

**Image outpainting.** Another practical application would be image outpainting that only draws the outer part of the image while keeping the input image the same. To do this, we experiment using the model trained on LSUN-Church and LSUN-Bedroom. First, we send images from the LSUN-Church validation dataset and LSUN-Bedroom validation dataset to the image encoder to obtain the global conditions. Then, to keep the original image the same, we replace the inner predicted noised image patches with the ground truth noised images patches (adding corresponding noise to the input images) during each timestep while sampling. We present our results in Figure 7. It can be seen from the results that our model can "imagine" the surrounding areas reasonably and generate rather consistent outer parts of the image without obvious border effects.

**Image inpainting.** In this task, we infill the corrupted images with random masks, which requires the restored results to be consistent in context. We experiment on the LSUN-Church validation set using already trained models without further tailored training. The original images are masked by different numbers of blocks ranging from 1 to 6, and the sampling process is only conditioned on local position embeddings w/o global conditions. The results are presented in Figure 8. From the figure, we can see that our model can infill the blocks consistently using surrounding patches, demonstrating that feature collage facilitates the model with the capability to be aware of adjacent information, enabling it to be naturally applied to inpainting tasks.

## 6 ABLATION STUDY

Three indispensable components: semantic code, position embeddings and shift window strategy on feature levels, considerably eliminate border artifacts and improve our model performance. Here,

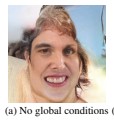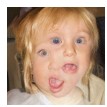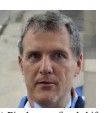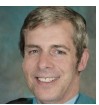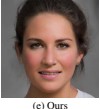

Figure 8: **Image Inpainting on LSUN-Church.** Six pairs of images are presented here. For each pair, the left side is the masked image; the right one is the inpainted result by our model without further training on this task. The number of masked patches increasing from left to right.

we conduct ablation study to investigate the effects of these modules. We provide both qualitative results and quantitative results in Figure 9 and Table 4 respectively.

**Global conditions.** We study the problem without global conditions; thus, the generation process will fully rely on the positional embedding and neighboring context information. We present our images in Figure 9 (a). It's interesting to see how the model generates when no global conditions are given, which is a strong constraint for the model to generate semantic-related patches. From the given image, we can see that the model can still generate locally-consistent images; the image quality is however relatively low.

**Position embeddings.** The last section shows that global conditions are necessary for our model to generate high-quality images. We then condition the model only on those to investigate the role of positional embeddings. The results are shown in Figure 9 (b). Without the position information, the model would generate distorted images with patchTheelonging to where they should be, although the whole image may follow a certain style. Hence, the positional embeddings are vital to our model.

**Collage in the pixel space.** A straightforward idea is to perform collage in the pixel space as present in Figure 2(b); the images are decomposed by window-shifted patches from their original positions. To maintain patch size consistency, we add zero padding around the image. For the sampling procedure: In an odd-number step, original patches are generated independently, while in an even-number step, patches with shifted positions are sampled.

Under this scheme, we experiment in two different settings. The first is to take a fixed shift step (half patch size) along the height and width direction. The sample result is shown in Figure 9(c). There are still apparent artifacts along the border. This proves that even though the shift window on the image level could enable patches to be aware of surroundings during sampling, the awareness level is quite limited, and the final generation is similar to breaking the image into smaller patches.

The second setting is to shift the patch position with a randomly sampled step ranging from zero to patch size. The inference result is shown in Figure 9(d). The sample quality is much improved compared to the previous situation. However, the result is still not as photo-realistic as Patch-DM. The reason is that although the random shift enables finer surrounding awareness, it lacks in-depth feature interaction as our model does. Therefore, the feature-level window shift and collage can significantly eliminate border artifacts and improve final inference quality.

Table 4: FID evaluation on 1,000 images of different ablation settings to investigate the importance of semantic condition, position embedding, and feature-level window shift.

| Method | FID (1k) ↓ |
|---|---|
| No global semantic condition | 79.33 |
| No position embedding | 48.82 |
| Pixel space fixed shift | 49.80 |
| Pixel space random shift | 52.11 |
| Patch-DM (Ours) | **37.99** |

(a) No global conditions (b) No position embeddings (c) Pixel space fixed shift (d) Pixel space random shift (e) Ours

Figure 9: Ablation study on global conditions (a), position embeddings (b), and feature level shift (c, d).

# 7 CONCLUSION

We have presented a new algorithm, Patch-DM, a patch-based denoising diffusion model for generating high-resolution images. We introduce a feature collage strategy to combat the boundary effect for patch-based image synthesis. Patch-DM achieves a significant reduction in model size and training complexity compared to the standard diffusion models trained on the original size images. Competitive quantitative and qualitative results are obtained for Patch-DM when trained on several image datasets.

ACKNOWLEDGMENTS

This work is supported by NSF Award IIS-2127544 and IIS-2211258.

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

## A    MODEL HYPERPARAMETERS

For model architecture, we base our diffusion model from Dhariwal & Nichol (2021) with changes of taking global semantic condition and positional embedding. The hyperparameters for the main denoising U-Net model are specified in Table 5. Since the model is resolution agnostic, the main architectures for all datasets keep the same. We adopt two methods to obtain global semantic conditions: for relatively low-resolution images, an encoder is trained with architecture borrowed from the first half of the U-Net model, and the architecture details of the encoder are shown in Table 6. For high-resolution images such as 1024×512, a pretrained image encoder is used to avoid scaling up the overall model size. We use ViT-B/16 in CLIP to obtain the image embeddings and optimize them during training. For position embeddings, we use sinusoidal positional embeddings. Time embedding and positional embedding are concatenated and modulated into ResBlocks together with the global code.

For realizing unconditional image synthesis, a latent diffusion model is trained on semantic embeddings. The implementation is based on the one proposed in Preechakul et al. (2022) with MLP + skip connections architecture. The parameter details are specified in Table 7.

## B    MORE QUALITATIVE RESULTS

We provide more unconditional sampling results with the models trained on our self-collected nature images (1024×512) in Figure 10-12. We also provide results on LHQ(1024×1024) and FFHQ(1024×1024) in Figures 14 and 13 respectively as well as three other standard benchmarks with a resolution of 256×256: LSUN-Bedroom (Figure 15), LSUN-Church (Figure 16), and FFHQ (Figure 17).

| Parameter | Patch-DM |
|---|---|
| Patch input size | $3\times64\times64$ |
| Channel multiplier | [1, 2, 4, 8] |
| Net channel | 64 |
| ResBlock number | 2 |
| Attention resolution | 16 |
| Batch size | 16 |
| Diffusion steps | 1000 |
| Noise scheduler | Linear |
| Learning rate | 0.0001 |
| Optimizer | Adam |

Table 5: Model Architecture for diffusion model.

| Parameter | Semantic Encoder |
|---|---|
| Input size | $3\times256\times256$ |
| Channel multiplier | [1, 2, 4, 8, 8] |
| Net channel | 64 |
| ResBlock number | 2 |
| Attention resolution | 16 |
| Global condition dimension | 512 |
| Batch size | 16 |
| Learning rate | 0.0001 |
| Optimizer | Adam |

Table 6: Model Architecture for image semantic encoder.

| Parameter | Latent Diffusion |
|---|---|
| Input size | 512 |
| MLP layers | 10 |
| MLP hidden size | 2048 |
| Noise scheduler | Constant 0.008 |
| Batch size | 256 |
| Learning rate | 0.0001 |
| Optimizer | Adam (weight decay 0.01) |

Table 7: Model Architecture for latent diffusion model.

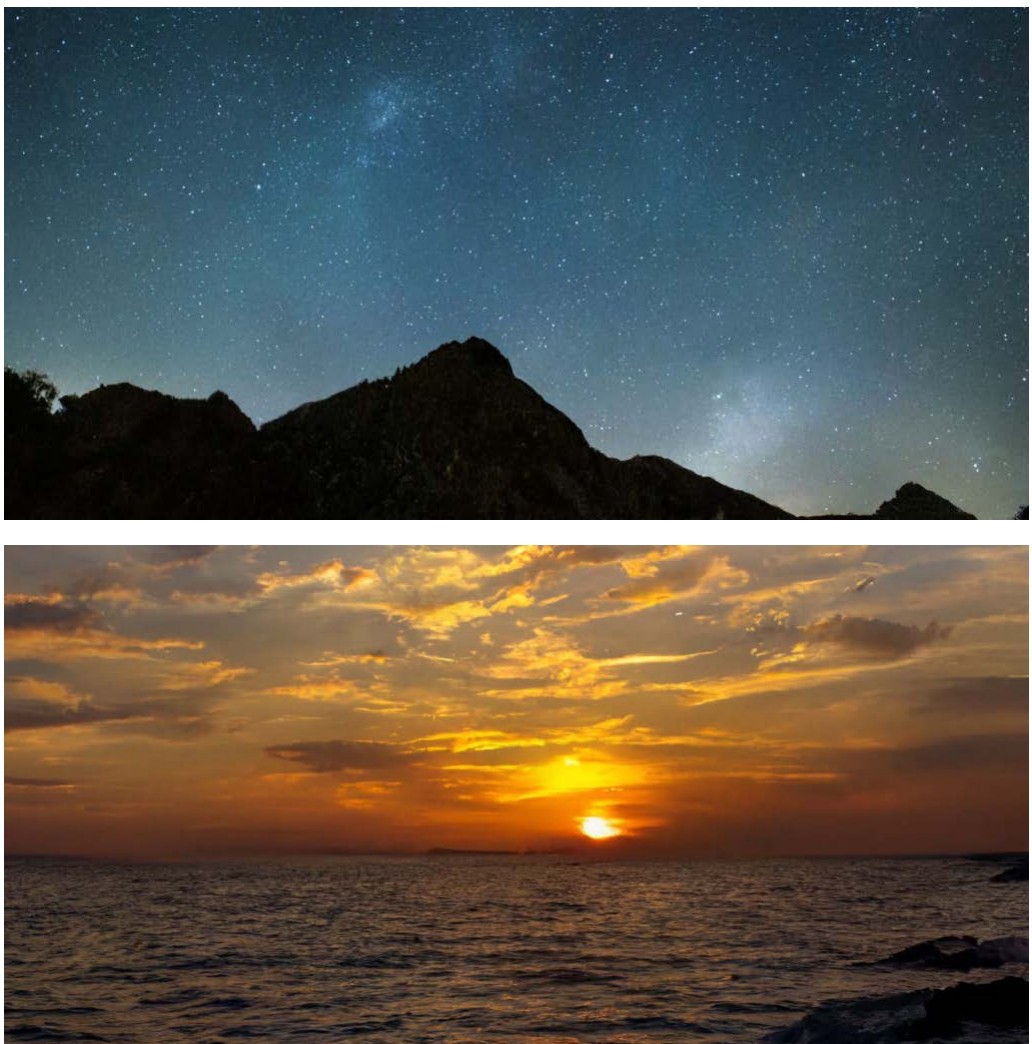

Figure 10: Additional qualitative results on self-collected nature dataset (1024×512).

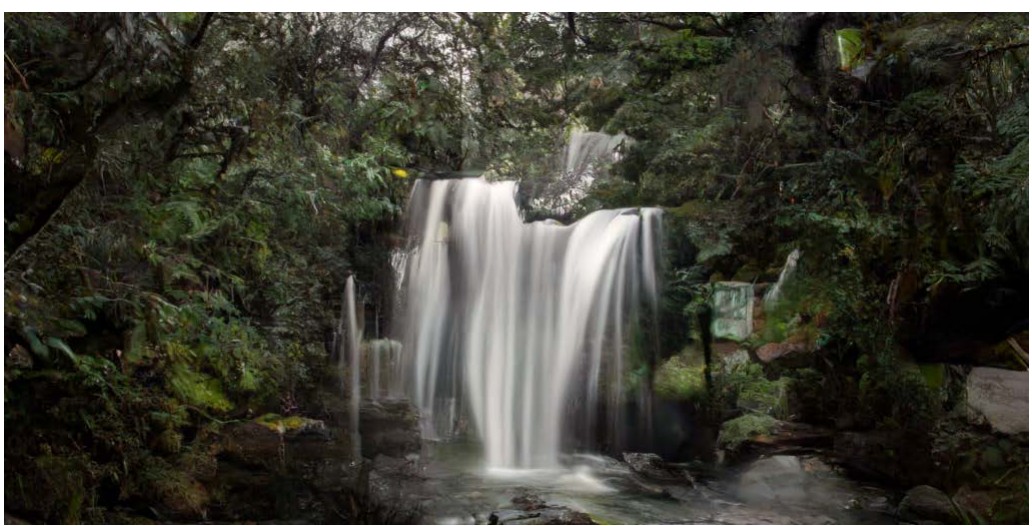

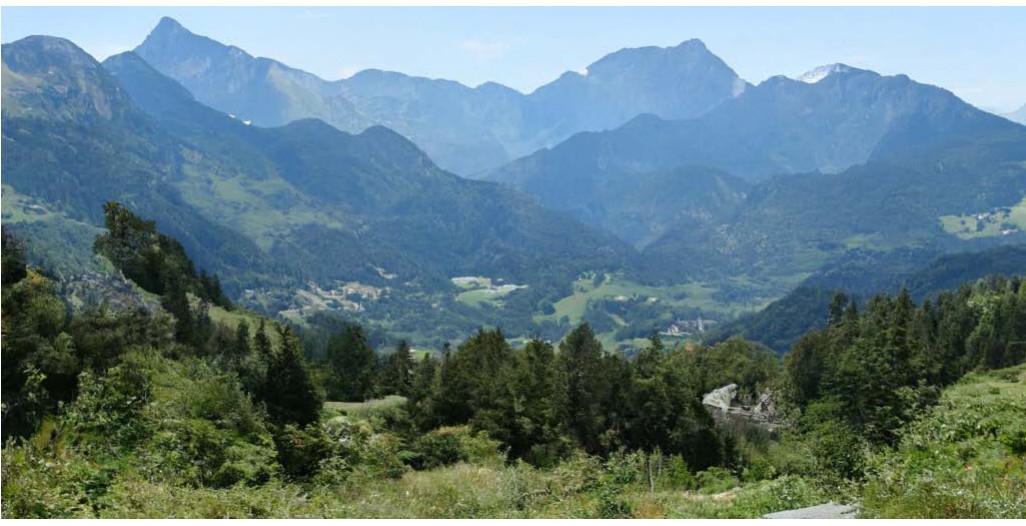

Figure 11: Additional qualitative results on self-collected nature dataset (1024×512).

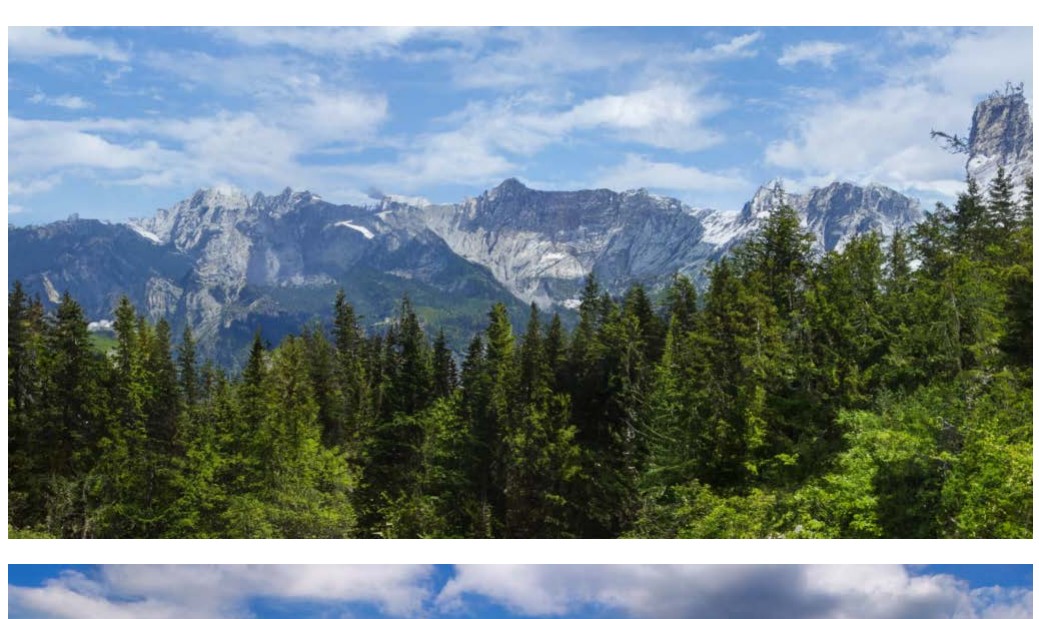

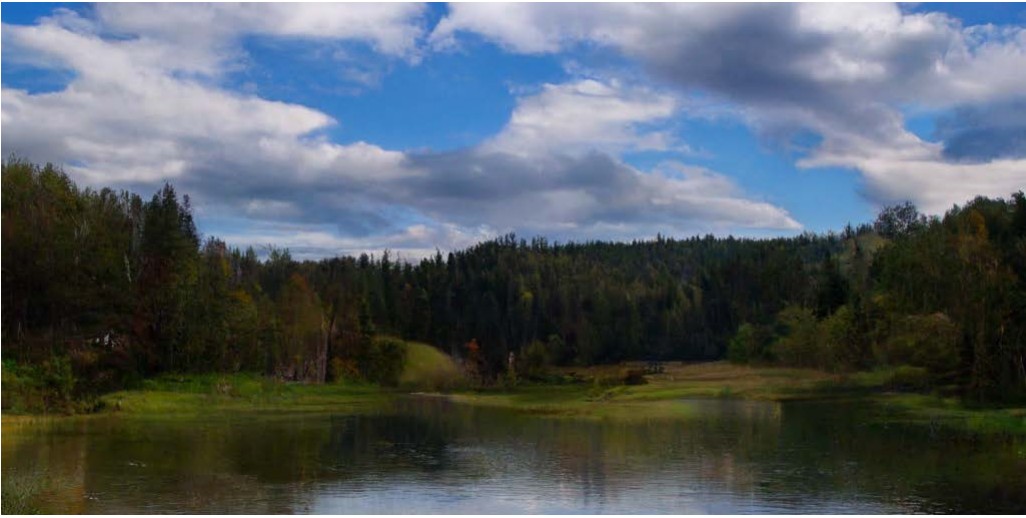

Figure 12: Additional qualitative results on self-collected nature dataset (1024×512).

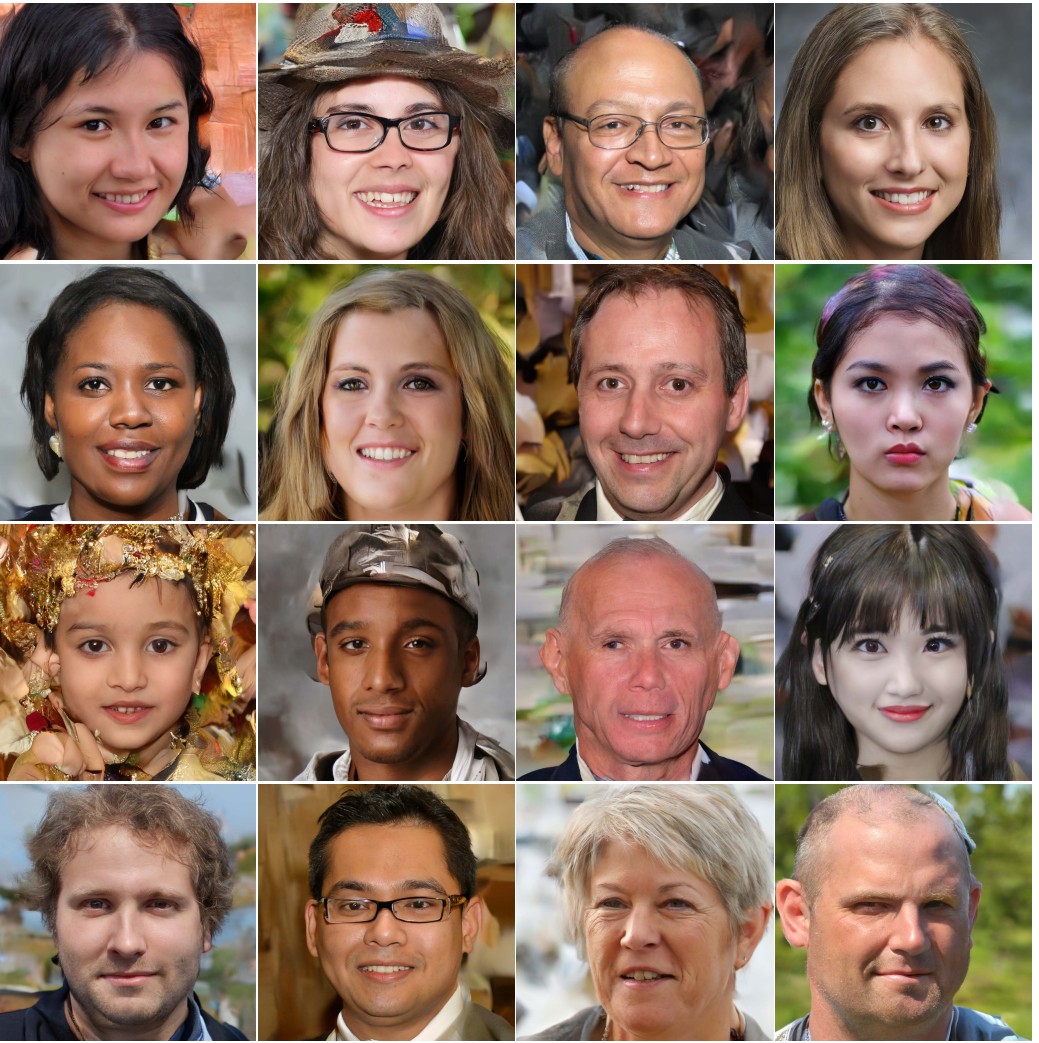

Figure 13: Additional qualitative results on high resolution FFHQ dataset (1024×1024).

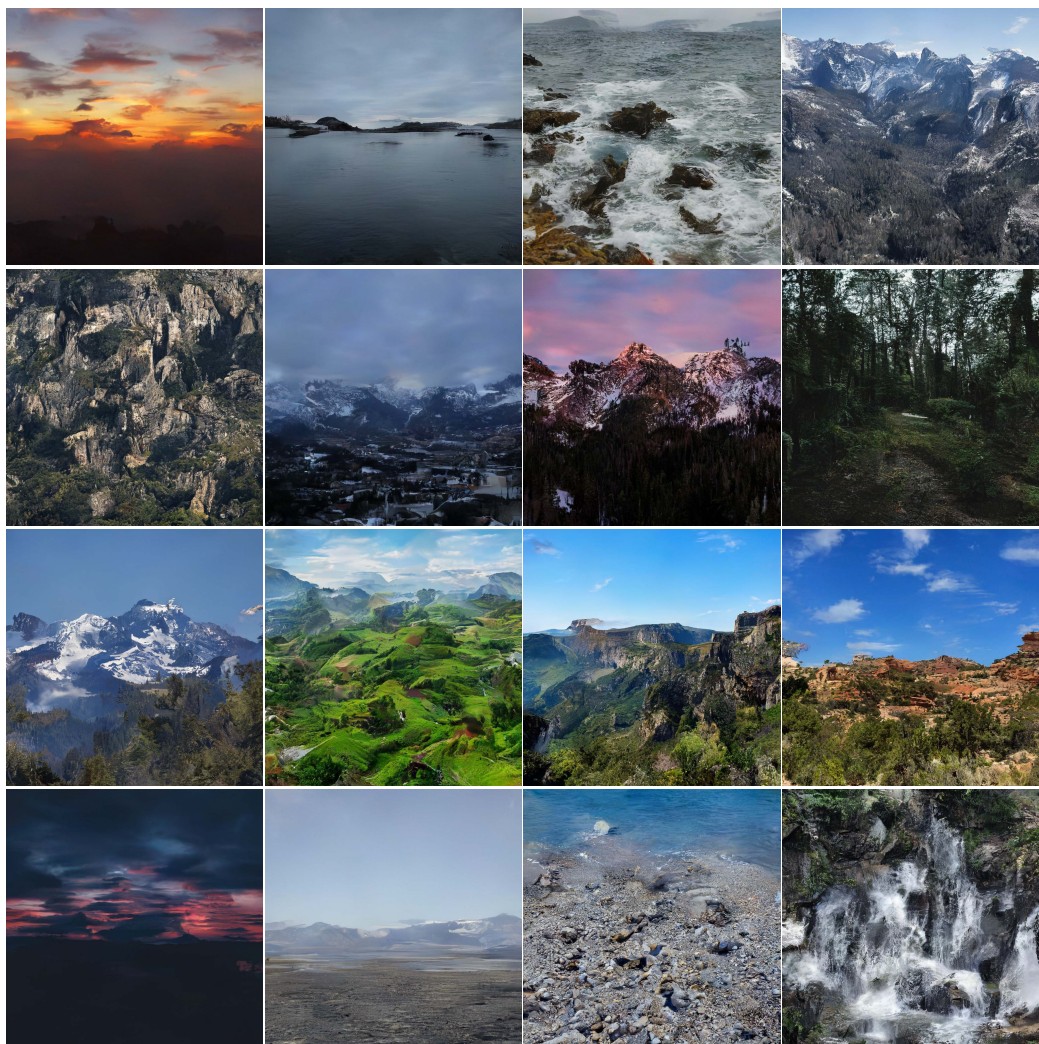

Figure 14: Additional qualitative results on high resolution LHQ dataset (1024×1024).

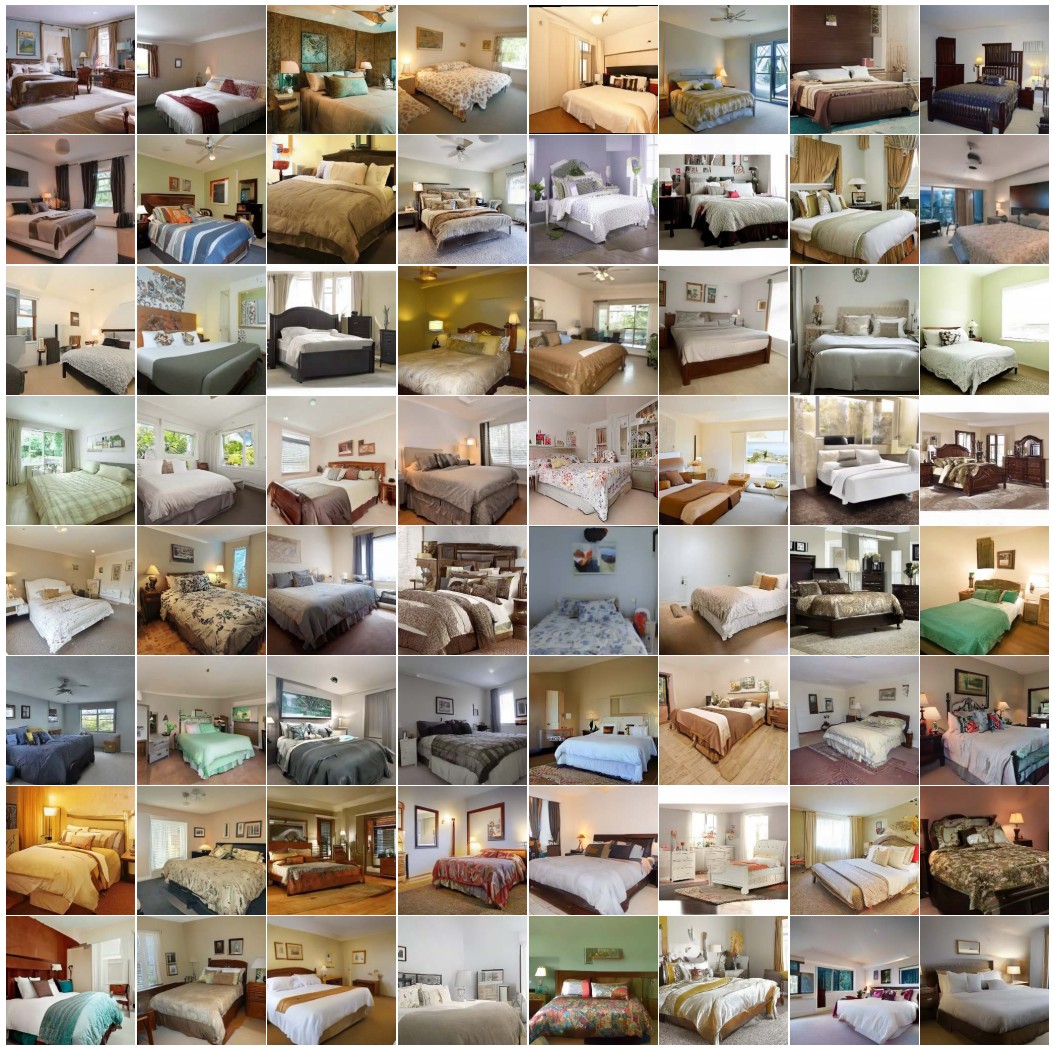

Figure 15: Additional qualitative results on LSUN-Bedroom (256×256).

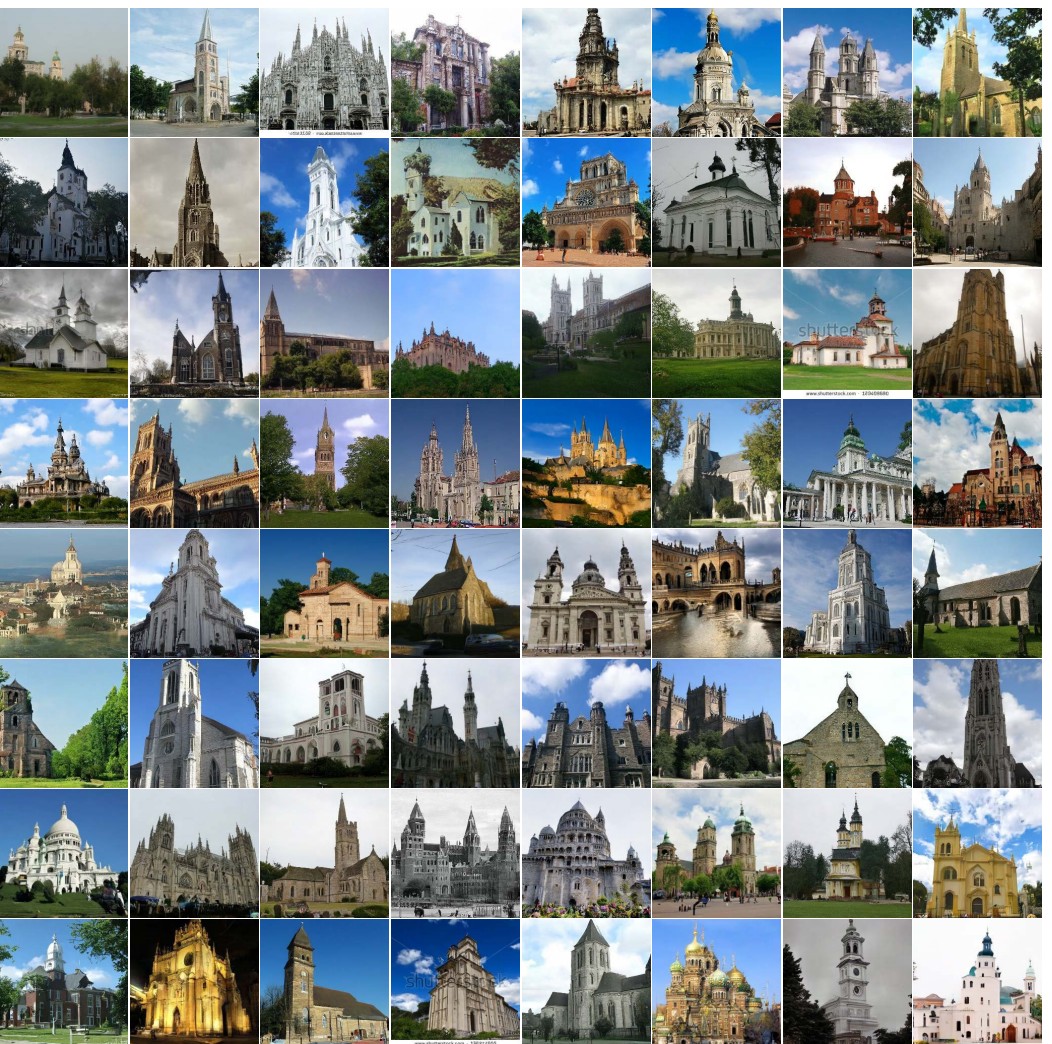

Figure 16: Additional qualitative results on LSUN-Church (256×256).

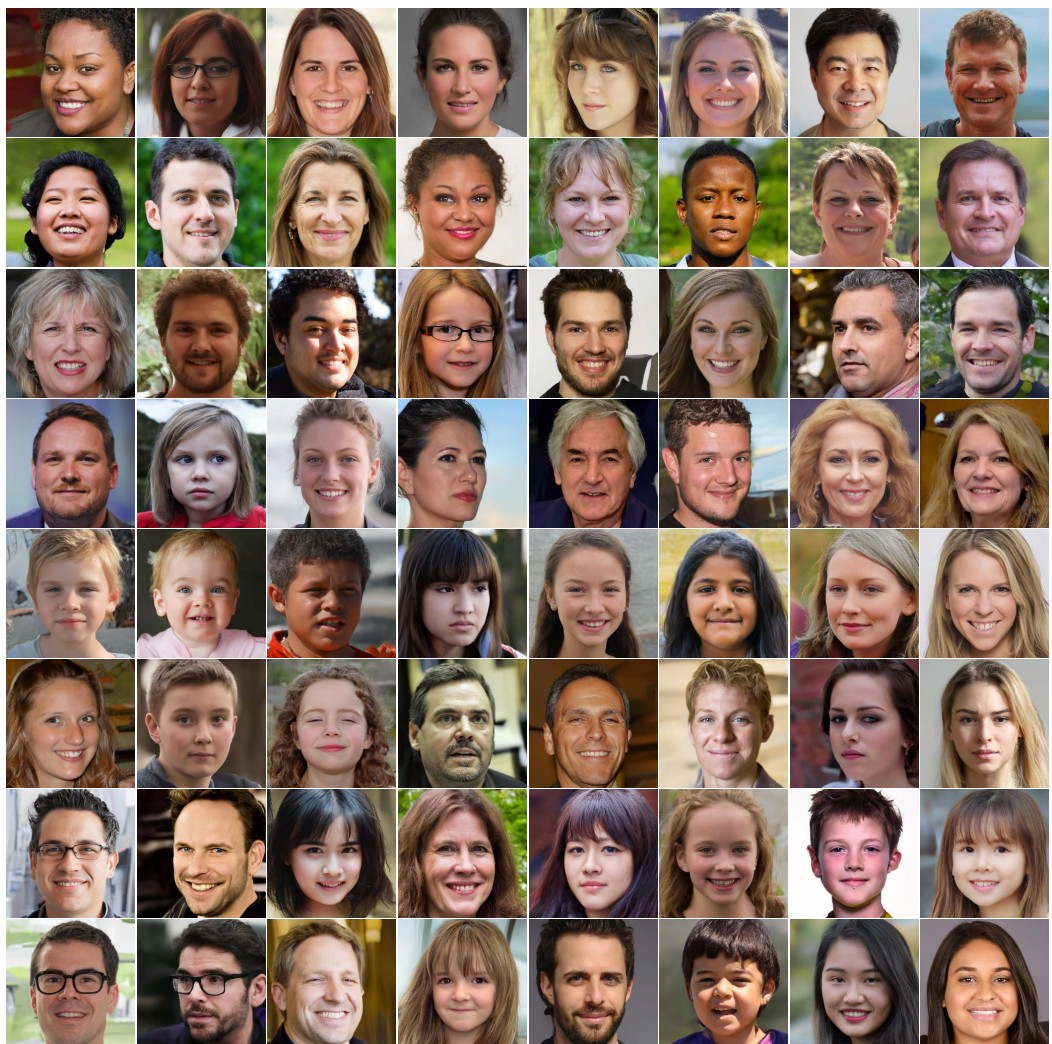

Figure 17: Additional qualitative results on FFHQ (256×256).

## C FAILED CASES COMPARISON ON FFHQ1024

To provide a more thorough understanding of the limitations of our proposed method, we compare our FFHQ1024 results with other non-patch-based methods. We use FFHQ1024 as it's a more structural dataset while other landscape datasets are not that structural, thus FFHQ1024 could better show how well the models learn the structural information. As there lacks results for FFHQ1024 using diffusion models, we compare our results with two state-of-the-art GAN-based methods on FFHQ1024: StyleGAN3(Karras et al., 2021) and StyleGAN-XL(Sauer et al., 2022) in Figure 18. It can be seen that non-patch GAN-based methods generally perform well on global consistency while ours may not perform well in such cases. For example, in the first row of our results, the wrinkles around the mouth are asymmetrical. And in the second row of our results, the eyes/eyeglasses are asymmetrical. As our method uses a patch size of $64 \times 64$ and there are a total of $16 \times 16 = 256$ patches, the global consistency sometimes may not perform well. We think one can further improve this by utilizing a larger patch size or introducing better global-consistency-enforcing mechanisms which we leave for future work.

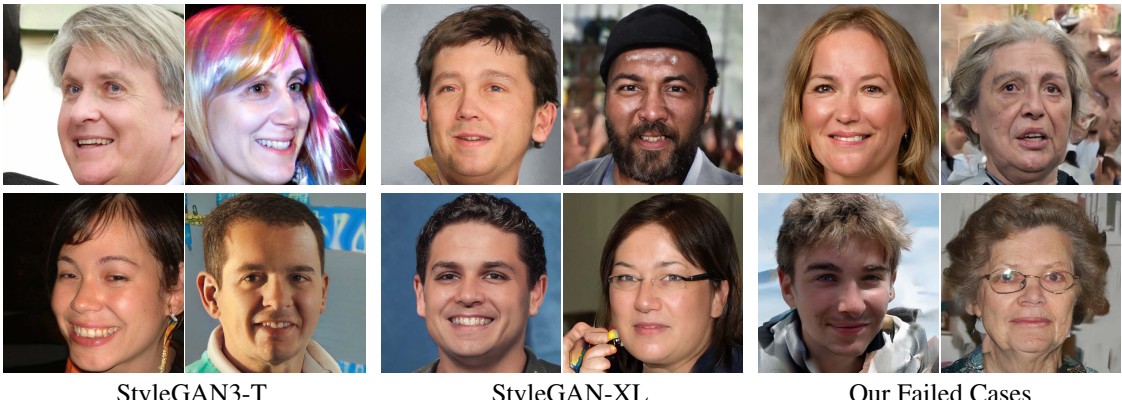

| StyleGAN3-T | StyleGAN-XL | Our Failed Cases |

Figure 18: Comparison on FFHQ1024 with StyleGAN3-T, StyleGAN-XL and our failed cases. Ours might fail on global consistency such as the wrinkles around the mouth in the first row and the eyes/glasses in the second row.

