# OpenReview forum: "Patched Denoising Diffusion Models For High-Resolution Image Synthesis"
_ICLR.cc/2024/Conference — ICLR 2024 poster_

### Official Review · Reviewer_oyqb · 2023-10-26

**Soundness:** 3 good
**Presentation:** 3 good
**Contribution:** 2 fair
**Rating:** 5
**Confidence:** 4

**Summary:**

This paper proposes a new feature collage strategy for the generative diffusion model to avoid boundary artifacts when synthesizing large-size images, termed Patch-DM. Feature collage systematically crops and combines partial features of the neighboring patches to predict the features of a shifted image patch, allowing the seamless generation of the entire image due to the overlap in the patch feature space. Experiments reveal the superiority of 1K resolution generation results on several datasets with 64×64 patches.

**Strengths:**

This paper proposes a new feature collage strategy for generative diffusion model to avoid boundary artifact when synthesizing large-size images.

**Weaknesses:**

1.	The novelty is relatively small and the impact of this paper may be limited since the proposed method only focuses on the boundary artifacts produced by the patch collage.
2.	Lacking comparisons with aggregation sampling strategies proposed in StableSR [1], the sampling strategy in StableSR can be performed without more parameters and training.
3.	The experiments only measure models with FID, lacking results measured under other metrics, e.g., CLIPScore.
4.	Figures 5 and 6 look confusing. (Which pictures are from which datasets or methods seem to be unclear)

[1] Wang, J., Yue, Z., Zhou, S., Chan, K. C., & Loy, C. C. (2023). Exploiting Diffusion Prior for Real-World Image Super-Resolution. arXiv preprint arXiv:2305.07015.

**Questions:**

Can the model be performed to generate images of more than 1K or arbitrary resolutions?

---

> ### Author Response · Authors · 2023-11-22
> **Response to Reviewer oyqb**
>
> We thank reviewer oyqb for the valuable feedback.
>
> **The novelty is relatively small and the impact of this paper may be limited since the proposed method only focuses on the boundary artifacts produced by the patch collage.**
>
> The main motivation of ours is to build a patch-based diffusion model for high-resolution image synthesis as diffusion models are hard to directly optimize in image space and previous diffusion methods working on high-resolution image generation usually utilize a VQ-GAN to encode the image to latent space(Stable Diffusion) or a super-resolution model that upsamples the generated image to a higher resolution(DALLE-2, Imagen). Patch-based image generation has two main challenges, one is global consistency and the other is the boundary artifacts between patches. We utilize a global embedding for global image consistency and propose the mechanism that does the patch collage in the feature space to solve the boundary artifacts problem which we believe could benefit the community by providing another solution for generating high-resolution images using diffusion models (also more efficient as it could utilize a small model e.g., 64x64 to generate high-resolution images e.g., 1024x1024).
>
> **Lacking comparisons with aggregation sampling strategies proposed in StableSR [1], the sampling strategy in StableSR can be performed without more parameters and training.**
>
> Thanks for raising this paper to us. We’re not aware of this paper which was posted in May this year on Arxiv. There are some major differences between ours and StableSR. First, StableSR is a paper for the image super-resolution task while ours is to generate high-resolution images directly. Second, StableSR adds another module to the pretrained Stable Diffusion model and further train this added module for the image super-resolution task while ours train a patch-based diffusion model from scratch. Third, the aggregation sampling strategies utilize overlapping patches over pretrained stable diffusion models which train on complete images, with a feature map of the low-resolution input but our model does both training and inference on the partial image patches without such structured feature maps. We attempted to perform aggregation sampling but found the results to be blur e.g., an average blurred face. We assume the reasons behind this are that we trained on partial patch images and there are no structured feature maps to serve as conditions so the nearby patches will be largely different, unlike the ones with structured feature maps as conditions that can enforce the overall content and structure to be consistent.
>
> **The experiments only measure models with FID, lacking results measured under other metrics, e.g., CLIPScore.**
>
> Thanks for the suggestion. However, our paper focuses on unconditional image generation. We also measure the models with sFID besides the most used FID in the unconditional image generation domain. CLIPScore might not be a direct fit for us as it’s a metric for measuring the consistency of the generated images with the given texts or images which we don’t utilize for our unconditional generation.
>
> **Figures 5 and 6 look confusing. (Which pictures are from which datasets or methods seem to be unclear)**
>
> Thanks. In Figure 5, we label the pictures from the left to right as FFHQ, LSUN-Bedroom and LSUN-Church and they’re all generated by our method. In Figure 6, we utilize two datasets here: LSUN-Bedroom and LSUN-Church where the left half of each group is LSUN-Bedroom and the right half is LSUN-Church. The methods have been labeled accordingly as COCO-GAN, InfinityGAN and ours. We’ve updated the captions to add more clarifications on this.
>
> **Can the model be performed to generate images of more than 1K or arbitrary resolutions?**
>
> In our paper, we show an example of generating a 2048x1024 image in Figure 4 by adding more patches during inference time when only trains on 1024x512 images. Also, our method can be utilized to train on higher-resolutions by adding more patches or increasing the patch size.

---

### Official Review · Reviewer_Tdt5 · 2023-10-28

**Soundness:** 3 good
**Presentation:** 2 fair
**Contribution:** 2 fair
**Rating:** 6
**Confidence:** 2

**Summary:**

This work aims to resolve the limitation of the existing diffusion models towards generating high-resolution images. To this end, the authors present the ideas of feature collage, position embedding, and global conditioning to develop a unified approach, namely Patch-DM, to generate high-resolution images. Moreover, potential applications of outpainting and inpainting are also demonstrated. Comparisons on low-resolution images are conducted to reveal that the proposed method doesn't underperform by a large margin compared to other methods. Comparison of high-resolution images reveals that the proposed method achieves state-of-the-art results.

**Strengths:**

1. The ideas of feature collage, position embedding, and global conditioning and their impact on image generation is interesting.
2. The proposed method achieves state-of-the-art results on high-resolution image synthesis

**Weaknesses:**

1. On Low resolution image synthesis as in Table 2, the proposed method is not comparable to that of state-of-the-art.
2. Some of the implementation details, reasoning behind choices, are missing in the description. Please refer to the comments under Questions for details.

**Questions:**

1. Section 5.2: Since every image is segmented into smaller patches, the total number of model parameters is much smaller than other large diffusion models. => The computations might be lesser, why should the number of model parameters be lesser? Authors argue that they use light weight models, but it’s unclear what architectural changes they made for it as compared to existing diffusion models, and how those changes are justifiable. In fact, authors might be able to generate better quality images with heavier architectures and make Table 2 performance comparable to that of previous diffusion models.

2. Beyond patch generation: The first one is to add patches inside the original images so that the generated images can have a 2× resolution compared to the ones in the training dataset. => Why do we need to add patches to original images? During test time the images are supposed to be generated from random noise and hence there is no need for adding patches to original images. This sentence is confusing and should be rewritten for clarity. Also the position embedding adaptations is not well explained, so is the reason for different choices with respect to the two methods mentioned under this category.

3. Image inpainting: No details of the position embedding and the related changes is mentioned in this case.

4. Image inpainting: The details could be incomplete. It is unclear how the original contents in the non-masked region is maintained in the output. Why is the global embedding not used?

Minor Fix:
- Section 5.2, still be ”recovered” during => still be ``recovered'' during

---

> ### Author Response · Authors · 2023-11-22
> **Response to Reviewer Tdt5**
>
> We thank reviewer Tdt5 for the valuable feedback.
>
> **On low-resolution image synthesis as in Table 2, the proposed method is not comparable to that of state-of-the-art.**
>
> Thanks for raising this concern. The main motivation of ours is to build a patch-based diffusion model for high-resolution image synthesis as diffusion models are hard to directly optimize in image space and previous diffusion methods working on high-resolution image generation usually utilize a VQ-GAN to encode the image to latent space(Stable Diffusion) or a super-resolution model that upsamples the generated image to a higher resolution. Our method does not out-perform the state-of-the-art results on low-resolution image synthesis but still achieve good results both quantitatively and qualitatively. Nevertheless, our model contains much fewer parameters compared to the previous diffusion methods as shown in Table 2.
>
> On the other hand, our method outperforms all previous patch-based methods. We believe patch-based methods won’t have advantages on low-resolution images compared to non-patch-based methods as the current computation resources for these tasks are sufficient. But our method could offer potential values for high-resolution image generation as using diffusion models to directly generate high-resolution images is still a hard problem which our method avoids optimizing on high-resolution images spaces by utilizing a patch-based method with Feature Space Patch Collage.
>
> **Section 5.2: ...The computations might be lesser, why should the number of model parameters be lesser? ... In fact, authors might be able to generate better quality images with heavier architectures and make Table 2 performance comparable to that of previous diffusion models.**
>
> The previous methods on diffusion models such as the well-known ADM usually utilize different architectures for different resolutions. As our method takes 64x64 as input, we utilize the 64x64 model architecture which has much fewer parameters than the 256x256 model. Thus when generating 256x256 images, our model has fewer parameters than other non-patch-based diffusion models. Yes, we agree that with larger model size, ours might offer better performance due to the increase of model capability. We didn’t try it as it’s not our main focus. But it might be a good future direction as we could enlarge the model or patch size to gain much better image quality. Here we mostly would like to demonstrate that we could utilize a smaller model (e.g., for 64x64 image generation) to generate good quality higher-resolution images (e.g., 256x256 or 1k-resolution).
>
> **Beyond patch generation: ...Why do we need to add patches to original images? During test time the images are supposed to be generated from random noise ... Also the position embedding adaptations is not well explained, so is the reason for different choices with respect to the two methods mentioned under this category.**
>
> When we generate 512x1024-resolution images same as the training dataset, we need 8x16 patches (random gaussian noises) to start with. By adding more patches we mean that if we’d like to generate a 2x resolution image, we need another 8x16 patches (random gaussian noise maps) to do the generation leading to a total patch number of 16x32. Regarding the position embedding, the reason we chose differently is based on the following intuition: when adding patches internally, the interpolated positional embedding can offer position information. However, when adding patches outside, the positional embeddings could not be interpolated thus we chose to not use position embedding.  We’ve updated the manuscript to clarify more on this. Thanks for raising this confusion.
>
> **Image inpainting: No details of the position embedding and the related changes is mentioned in this case. The details could be incomplete. It is unclear how the original contents in the non-masked region is maintained in the output. Why is the global embedding not used?**
>
> For the image inpainting, we use the position embedding but no global embedding is used. In each diffusion step, the non-masked region is replaced by the input image (with noise added). Thus the original contents in the non-masked region are maintained in the output. The global embedding is not used since we don’t have the complete image to obtain the global embedding. Therefore, in the image inpainting application, we fully rely on the nearby image content and the position embedding to recover the missing image content.

---

### Official Review · Reviewer_3rts · 2023-10-31

**Soundness:** 4 excellent
**Presentation:** 4 excellent
**Contribution:** 3 good
**Rating:** 5
**Confidence:** 4

**Summary:**

This paper proposes a patch-based denoising diffusion model called Patch-DM for generating high-resolution images. The key contributions are: 1. Introduces a feature collage strategy to avoid boundary artifacts when synthesizing images from patches. It combines partial features from shifted patches to predict features for a new patch. 2. Achieves state-of-the-art FID scores on 1024x512 natural images and 1024x1024 LSUN/FFHQ images using a lightweight model. 3. Demonstrates Patch-DM can directly generate high-fidelity 1K resolution images with minimal patch boundary effects. 4.Reduces memory complexity compared to full-image diffusion models for high-res synthesis. 5.Shows applications like image outpainting, inpainting, super-resolution without any post-training. 6. Validates through ablation studies that feature collage is better than pixel collage for spatial consistency. 7. Provides an effective patch-based generative modeling approach using diffusion models for high-resolution image synthesis with reduced costs.

**Strengths:**

•	Proposes Patch-DM, a novel patch-based denoising diffusion model that can generate high-resolution images directly without relying on hierarchical sampling. This simplifies the sampling procedure.
•	Introduces a new feature collage strategy to avoid boundary artifacts when synthesizing images from patches. It forces consistency by combining partial features from shifted patches.
•	Achieves state-of-the-art FID scores on generating natural images and LSUN/FFHQ images using a lightweight model, outperforming prior patch-based methods.
•	Qualitative results show Patch-DM can produce high-fidelity 1K resolution images with minimal patch boundary effects.

**Weaknesses:**

•	Based on my experience and recent related publications (e.g., "Weather Diffusion-PAMI'23"), patch-based diffusion models often lead to reduced inference efficiency. I hope the authors can provide specific comparisons of inference time and overall efficiency, especially compared to previous GAN methods.
•	The proposed method has limited technical contributions. The authors did not provide detailed explanations or theoretical justifications to explain why the Patch Collage in Feature Space strategy can avoid artifacts. Furthermore, the research and exploration of Semantic Code are not sufficiently in-depth.
•	The authors should provide a quantitative comparison of image inpainting and image outpainting results. Quantitative results would better demonstrate the superiority of the proposed method.

**Questions:**

•	Why does the Patch Collage in Feature Space strategy avoid artifacts? Can a detailed analysis and explanation be provided? This is crucial for future work.

•	What is the running speed of the proposed method? How much slower does the Patch Collage in Feature Space strategy make the model inference speed?

•	What are the limitations or further areas of exploration for the proposed method?

---

> ### Author Response · Authors · 2023-11-22
> **Response to Reviewer 3rts**
>
> We thank reviewer 3rts for the valuable feedback.
>
> **...I hope the authors can provide specific comparisons of inference time and overall efficiency, especially compared to previous GAN methods...**
>
> Thanks for the suggestions. We provide our inference efficiency compared with ADM and Diff-AE on 256x256 image generation below (all using NFE=50). As we utilize a 64x64 model to generate 256x256 images. Our model is much smaller and thus can have better efficiency.
>
> When compared to GAN methods (single-step), diffusion models won’t have advantages due to diffusion models’ iterative generation mechanism. For example, we did a test on StyleGAN3 of which the inference time is just 0.031s while most diffusion models will require several seconds or more on a single TITAN Xp GPU.  And ours is not exception as we’re developed based on diffusion models and requires multiple steps e.g.,50  to generate real images. Though GAN-based methods run considerably faster than diffusion-based methods, the exploding application of diffusion models suggests that the superior scalability and stability gain of diffusion models over GANs outweigh the compromise of the speed. Nevertheless, the main motivation of our method is trying to generate high-resolution images with small diffusion models. Right now diffusion models are hard to directly train on a high-resolution dataset while previous methods usually utilize a super-resolution way (Imagen or DALLE2) or latent-space way(Stable Diffusion) to generate high-resolution images. Patch-based methods might provide another direction for solving this problem which can directly generate high-resolution images by only operating on low-resolution patches.
>
> |                                  | GFLOPS | Inference time |
> |----------------------------------|--------|----------------|
> | ADM                              | 2227.5 | 17.77s         |
> | DiffAE                           | 2215.1 | 4.81s          |
> | Ours                        | 961.3  | 3.76s          |
>
> **..did not provide detailed explanations or theoretical justifications to explain why the Patch Collage in Feature Space strategy can avoid artifacts. Furthermore, the research and exploration of Semantic Code are not sufficiently in-depth. Why does the Patch Collage in Feature Space strategy avoid artifacts? Can a detailed analysis and explanation be provided? This is crucial for future work.**
>
> We provide ablations on using patch collage in pixel space in the ablation section of the paper(Figure 9 & Table 4) and show our method’s superiority over the pixel space. The main reason behind this is that it lacks in-depth feature interaction as patches can only interact with other patches in the input image level using the collage in the pixel space. In our method, as the patches can interact with other patches in deep feature levels, the interaction enables better surrounding awareness. Furthermore, patch collage in pixel space can only be done in the original image level while patch collage in feature space can be done in multiple feature levels as the network has many hidden features and they represent different information of the images.  On the other hand,  lots of works on computer vision such as FPN utilizes multiple levels of features from CNN to enable better semantic understanding. The interaction between different patches on multiple levels of features have similar effects here. The generation also requires good representation and the shifted patches in feature space will provide better representation than the shifted patches in pixel space.
>
> For the semantic code, we also provided ablations in the paper (Figure 9 & Table 4). The main reason for the semantic code is to enforce global awareness of the generated images since the patch collage mechanism in the feature space is more focused on making the nearby patches consistent which can be seen in Figure 9 where images lack global consistency if no semantic code is provided.

---

> ### Author Response · Authors · 2023-11-22
> **Response to Reviewer 3rts (Continued)**
>
> **The authors should provide a quantitative comparison of image inpainting and image outpainting results. Quantitative results would better demonstrate the superiority of the proposed method.**
>
> Thanks for the suggestion. For image inpainting, we compare our method with SDEdit which can use vanilla diffusion models for image inpainting. When it comes to image outpainting, as vanilla diffusion models that work on fixed-resolution images is hard to directly perform zero-shot outpainting, we compare our method with previous patch-based methods such as COCO-GAN and Inifinity-GAN. Notice that for outpainting with given input, as both COCO-GAN and Infinity-GAN needs to do GAN-inversion and cannot preserve the input faithfully, therefore we compare our method on loosely outpainting(no given input) which is to add patches to unconditionally generate larger images (e.g., generate 384x384 images while trained on 256x256 images) in a zero-shot manner. We provide all the results below. And we can see that our method outperforms the previous methods list in the table on inpainting and outpainting(no given input). On the other hand, our method can support all three tasks in a zero-shot manner. While vanilla diffusion models only support inpainting and previous patched GAN-based methods need to do GAN inversion for image outpainting and does not support inpainting in a zero-shot manner.
>
> |                                 | Inpainting |        | Outpainting|(w/ Given input)        | Outpainting|(w/o given input)         |
> |---------------------------------|------------|--------|-----------------------------|--------|------------------------------|--------|
> |                                 | bedroom    | church | bedroom                     | church | bedroom                      | church |
> | SDEdit(Vanilla Diffusion Model) | 42.67      | 40.85  | -                           | -      | -                            | -      |
> | COCOGAN                         | -          | -      | -                           | -      | 58.38                        | 86.48  |
> | InifinityGAN                    | -          | -      | -                           | -      | 55.01                        | 38.01  |
> | ours                            | **41.23**      | **39.32**  | 53.19                       | 55.49  | **48.73**                        | **31.05**  |
>
> “-” denotes the method does not support this task in a zero-shot manner. COCOGAN and ours both utilize a patch size of 64x64 while InifinityGAN utilizes a patch size of 101x101. We use the validation dataset for inpainting and outpainting(w/ given input). For outpainting(w/o given input), we randomly generate 1000 images.
>
> **What is the running speed of the proposed method? How much slower does the Patch Collage in Feature Space strategy make the model inference speed?**
>
> We compare the speed of the proposed method above and show that our method has advantages when generating 256x256 images over ADM and Diff-AE. The main reason is that we utilize a much smaller model. For Patch Collage in Feature Space strategy, though it needs to do split and merge operations on the vectors, it does not contain any numeric operations. To evaluate the effects on the efficiency of this operation, we did a test on this specific operation and found that it only accounts for 4% of total inference time.
>
> **What are the limitations or further areas of exploration for the proposed method?**
>
> The limitations of our method should be similar to other patch-based methods which might not be as good as non-patch-based methods in global consistency as they optimize the original pixel space directly. A simple solution to solve this might be to increase the patch size. For example, in our 1024x1024 experiments there are 16x16 patches but increasing the patch size could lead to fewer patches which could alleviate this problem as like in our 256x256 results where there are only 4x4 patches. For future exploration, please see our general response above.

---

### Official Review · Reviewer_7jCc · 2023-11-01

**Soundness:** 3 good
**Presentation:** 3 good
**Contribution:** 2 fair
**Rating:** 6
**Confidence:** 4

**Summary:**

The paper proposes a denoising diffusion model, Patch-DM, for generating high-resolution images (e.g., 1024×512), trained on small-size image patches (e.g., 64×64). The major contribution of the paper is a new feature collage strategy, which is designed to avoid the boundary artifact when synthesizing large-size images. The authors demonstrate the effectiveness of  Patch-DM on mage synthesis results on their newly collected dataset of nature images (1024×512), as well as on standard benchmarks of LHQ(1024× 1024), FFHQ(1024× 1024) and on other datasets with smaller sizes (256×256), including LSUN-Bedroom, LSUN-Church, and FFHQ. The show state-of-the-art FID scores on all six datasets for the proposed model. Further, Patch-DM also reduces memory complexity compared to the classic diffusion models.

**Strengths:**

1) The paper is reasonably well written and easy to follow
2)  The quantitative results demonstrated in Table 1 and Table 2 shows that the model outperforms state of the art.

**Weaknesses:**

1) The paper is not sufficiently novel. I'm not working in this domain, but the only novel part that the authors state is creating the collage of the patches in the feature / latent space based on their spatial embeddings. This does not sound like something that has not been done before in the field of image generation. It would  be helpful if the author come up with a more comprehensive literature survey that provides more related works to this particular  design choice and clearly shows the difference. For example, from a short search I found the following relevant paper: [1] https://arxiv.org/pdf/2207.04316.pdf --  Improving Diffusion Model Efficiency Through Patching
[2] https://arxiv.org/abs/2304.12526 -- Patch Diffusion: Faster and More Data-Efficient Training of Diffusion Models (note the paper was first submitted on April 2023).

2) Even if we consider the combination of Patch Diffusion in the latent space sufficiently novel, from the analysis in the supplementary material, I find that faces demonstrate usual artifacts around eyes and mouse (and I think that this is happening despite training on a dedicated dataset). Midjourney models generate much better faces. It would be great to understand why the proposed model fails on those.

**Questions:**

1) Can you please add comparison to other techniques in the supplementary? It might be useful to reduce the example to great examples vs. poor examples and provide some discussion on failure modes
2) In Table 2 - "We bold the numbers to denote the best numbers in the same category." --> can you please explain what you mean by "the same category"

---

> ### Author Response · Authors · 2023-11-22
> **Response to Reviewer 7jCc**
>
> We thank reviewer 7jCc for the valuable feedback.
>
> **The paper is not sufficiently novel... does not sound like something that has not been done before in the field of image generation. It would be helpful if the author come up with a more comprehensive literature survey...**
>
> Thanks for the suggestion. There have been works using GANs for patch-based image generation, such as COCO-GAN, Infinity-GAN, AnyresGAN which we list them in the related works. To the best of our knowledge, we’re not aware of such works that create the collage of the patches in the feature / latent space based on their spatial embeddings before, especially for diffusion models. We think the GAN-based methods (latent to image)  lack encoder architecture which makes it hard to create the collage of the patch features. Diffusion models have an image-to-image structure and also have an iterative mechanism which are suitable for our design.
>
> Currently, there are few works working on patch-based diffusion models. [1] does a reshaping operation on the input image which only pushes the dimensions of the height and width to the channels. More specifically, the reshaping operation is from [B C H W] to [B (Cxhxw) H/h W/w]. It’s not a true patch operation as the model still takes the whole image as input.  [2] does a patch operation during the training stage by concatenating another position embedding layer to the input. However, it still requires full-resolution operation during the inference stage. The paper doesn’t conduct experiments on high-resolution datasets, possibly due to this reason. On the other hand, our method performs patch operation in both training and inference stages which enables the high-resolution generation present in the paper. Besides these two, there are also other works applying patch-based diffusion models to specific applications like image restoration under weather conditions[3] and anomaly detection in brain MRI[4]. Both of them utilize a condition diffusion mechanism in which [3] patchifies weather-degraded images to serve as conditions while [4] attempts to recover a patch in the MRI image with the rest of the images as conditions to perform anomaly detection. Both of them are not able to perform image generation.  We’ve also updated the related works part of the manuscript.
>
> [1] Improving Diffusion Model Efficiency Through Patching\
> [2] Patch Diffusion: Faster and More Data-Efficient Training of Diffusion Models\
> [3] Restoring Vision in Adverse Weather Conditions With Patch-Based Denoising Diffusion Models\
> [4] Patched Diffusion Models for Unsupervised Anomaly Detection in Brain MRI
>
> **sufficiently novel... faces demonstrate usual artifacts around eyes and mouse... Midjourney models generate much better faces. It would be great to understand why the proposed model fails on those.**
>
> Thanks for pointing it out. We agree there might be some artifacts, though not significant, for the FFHQ1024 results. We think the main reason for this is we are using 64x64 patches to do 1024x1024 image generation, the patch number is 16x16 which might be a little larger. For comparison, COCO-GAN uses a 64x64-patch to train on 256x256 images, InfinityGAN uses 101x101-patch to train on 197x197 images. When training under the same settings, we can see from Table 1 that we outperformed all previous methods. When the setting comes to 64x64-patch on 256x256 training dataset where the patch number is 4x4 we can see the results usually don’t contain such artifacts. Some possible fixes might be to increase the patch size of 64x64 to 128x128/256x256 or add more layers to improve the capability of our model. We leave it for future work to further alleviate those minor artifacts in high-resolution structural datasets.
>
> On the other hand, midjourney is a large text-to-image diffusion model which trains on a very large dataset (no public technical report so we’re not aware of other details) which might not be a direct comparison.

---

> ### Author Response · Authors · 2023-11-22
> **Response to Reviewer 7jCc (Continued)**
>
> **Can you please add comparison to other techniques in the supplementary? It might be useful to reduce the example to great examples vs. poor examples and provide some discussion on failure modes.**
>
> Thanks for the suggestion. We’ve added a section in the revised supplementary material. We select FFHQ1024 as it's a more structural dataset while other landscape datasets are not that structural, thus FFHQ1024 could better show how well the models learn the structural information which is more challenging to generate high-resolution images.  As there lacks results on FFHQ1024 using diffusion models, we compare our FFHQ1024 results with state-of-the-art ones from non-patch GAN-based methods StyleGAN3 and StyleGAN-XL. Our results might not have as good global consistency as those results since we utilize 16x16=256 patches in our case. For example, the wrinkles and eyes/glasses might be  asymmetrical. We think one can further improve this by utilizing a larger patch size or introducing better global-consistency-enforcing mechanisms which we leave for future work.
>
>
> **In Table 2 - "We bold the numbers to denote the best numbers in the same category." --> can you please explain what you mean by "the same category"**
>
> In Table 2, we divide the methods into non-patch-based (top) and patch-based (bottom). We’ve updated the manuscript to make it more clear.

---

### Author Response · Authors · 2023-11-22
**Common Response on Contribution and Future Direction**

### **Contribution**
* Our motivation is to try to generate high-resolution images directly using diffusion models as previous diffusion models are hard to generate high-resolution images directly. They usually generate high-resolution images through super-resolution way(generate low-resolution images first and then upsample to high-resolution images) such as DALLE-2 or Imagen, or using a VQ-GAN to encode the images to latent space and train and inference in the latent space such as Stable Diffusion.
* Our method provides a direct way to generate high-resolution images using a very small model. For example, our results (1024x1024) are generated using a 64x64 diffusion model architecture and during both the training and inference stage, the model only needs to process 64x64 patches which makes it possible to generate high-resolution images on more affordable hardware.
* Our proposed Patch-DM, in which we introduce a Feature Space Patch Collage mechanism along with the global code and position embeddings to enforce the global consistency and the local(nearby features) consistency. The Feature Space Patch Collage could significantly alleviate the border artifacts and make the images more coherent. It performs much better than simply doing the collage operation in the pixel space as it supports multi-level in-depth feature interaction rather than one-level image interaction.
* We believe our method could benefit the community by providing another solution for generating high-resolution images using diffusion models (also more efficient as it could utilize a small model e.g., 64x64 to generate high-resolution images e.g., 1024x1024).

### **Future Direction**

We think there might be some future directions of our works.
* The first is that our proposed method could be utilized for high-resolution text-to-image generation that does not require a VAE model to encode the images to a smaller latent vectors like Stable Diffusion did or require a super-resolution model to upsample the image from low-resolution to high-resolution like DALLE2 or Imagen did. Other conditional image generation could also be a potential fit.
* Furthermore, as the proposed method is optimized for a single patch which might be able to go beyond traditional image generation to synthesize higher-resolution images than the training dataset. We’ve shown some preliminary results in the paper on 2x generation by adding more patches with interpolated positional embeddings.
* What’s more, pretrained diffusion models have been utilized for many other downstream tasks such as image editing and solving inverse problems but they sometimes require the model to directly take images as inputs since there exists some operations directly on images or external modules that take images as inputs. For example, image inpainting needs to replace the un-masked region with known image pixels and methods that solve general inversion problems using diffusion models like DDRM[1], DPS[2] need the guidance directly on the images. Our diffusion model directly takes images as input thus it would be ideal for such kinds of applications.
* Finally, regarding the efficiency, as our model can utilize a small model to generate high-resolution images and the model only needs to operate on the small patch, our model could allow for high-resolution generation using diffusion models on more affordable hardware.


[1] Denoising Diffusion Restoration Models\
[2] Diffusion Posterior Sampling for General Noisy Inverse Problems

---

### Meta-Review · Area_Chair_cHTR · 2023-12-10

**Metareview:**

This paper presents a way to generate high-resolution images from a diffusion model that was originally trained to generate low-resolution images. The high-resolution image is generated through image collage at feature space instead of pixel space.

Strengths in terms of high-resolution image synthesis generation quality (FID) and the simple inference process are recognized in the reviews.
At the same time, the reviewers raised concerns about
1) weak novelty in the collage idea
2) lack of comparisons with alternative methods
3) no CLIP score in evaluation metrics
4) worse FID for lower-resolution image synthesis (256x256)
5) justification for adding patches to the original images during training
6) inference efficiency comparisons
7) lack of analysis on for why artifacts are avoided in the collage process
8) lack of in-depth exploration on Semantic Code
9) quantitative comparisons for image outpainting task

In the rebuttal, the authors provided answers to the questions.
1) While there are several works out there using similar ideas of collage, this method avoids the necessity of high-resolution dedicated models. It is practically useful.
2) The suggested methods is on arXiv but wasn't published in peer-reviewed publications.
3) CLIP score is not necessary as the experiments are focusing on unconditional image generation.
4) 256x256 result is not as good as SoTA methods but the model is smaller. Limitations to be discussed.
5) answered & manuscript updated
6) presented with efficiency proof
7) provided in the manuscript
8) provided in the manuscript
9) provided in the rebuttal

While several of the concerns are valid, the authors addressed most of them and the practical gains of the patch-DM outweigh the limitations. I would recommend the acceptance of this paper. The authors are recommended to reflect the updates in the rebuttal in the final manuscript.

**Justification For Why Not Higher Score:**

Although most of the concerns from the reviewers are addressed, exposition could still be improved and limitations should be better discussed in the manuscript.

**Justification For Why Not Lower Score:**

Whilst there is some room to improve, most of the concerns are resolved and the practical gains from patched image synthesis to generate high-resolution images from a low-resolution diffusion model outweigh them.

---

### Decision · Program_Chairs · 2024-01-16

Accept (poster)